# Popular Catholicism Puerto Rican Style: The Virgin of Rincón, Human Agency, and Miracles

Angel D. Santiago-Vendrell

George W. Truett Theological Seminary, Baylor University, Waco, TX 76657, USA; angel_d_vendrell@baylor.edu

**Abstract:** In the past, popular Catholicism in Latin America and the Caribbean was perceived with suspicion by liberation theologians and official Roman Catholicism for its eccentricities, lack of doctrinal coherence, and fears of syncretism with folk religions. Nowadays, popular Catholicism in Latin America and the Caribbean has been a source of theological reflection, ecumenism, and religious revitalization. The apparition of the Holy Mother in 1953 at barrio Rincón in Sabana Grande, Puerto Rico, is a case study in global Catholicism that exemplifies this turn to see popular Catholicism as a source of liberation, perseverance, and deep spiritual devotion by the faithful. Using cultural, social, and reception historiography, the article argues that the Puerto Rican faithful were not passive recipients of the literary narratives of journalists covering the events as narrated by the main protagonists, the children/seers, but rather themselves formulators of history through their reception and participation. This is demonstrated by the allegiances of the faithful to popular Catholicism and their rejection of the official mandates of the clergy to ignore the events taking place at barrio Rincón regarding the apparition of the Virgin.

**Keywords:** popular Catholicism; miracles; Virgin Mary; devotional spirituality





## 1. Introduction

There was a time in the history of Latin American Christianity that popular Catholicism was perceived as a deviation of official or orthodox forms of Christianity. In Roman Catholic statements, syncretism is usually associated with the religion of the people or popular religion. The second conference of Latin American bishops claimed in the final document that, in the past, "Due to lack of attention on the part of pastoral agents and to other complicated factors, the religion of the people had been criticized as signs of erosion and distortion with aberrant substitutes and regressive forms of syncretism" (Eagleson and Scharper 1979, p. 186). Liberation theologian Juan Luis Segundo who was no fan of popular religion described it as syncretistic, superstitious, simplistic, and in need of re-evangelization (Segundo 1976, p. 183). Even though today, popular Catholicism is a source of liberation, cultural identity, and spirituality, there continues to be elements that need purification. As Pope Benedict XVI claimed in the inaugural sermon of the Fifth General Conference of Bishops in Latin America and the Caribbean, "This religiosity is also expressed in devotion to the saints with their patronal feasts, in love for the Pope and the other Pastors, and in love for the universal church as the great family of God, that neither can nor ever leave her children alone or destitute. All this forms the great mosaic of popular piety which is the precious treasure of the Catholic Church in Latin America, and must be protected, promoted and, when necessary, *purified*" (Pope Benedict XVI 2007).

This article will follow the insights and methods of social and cultural historians and *rezeptionsgeschichte* historiography to accomplish this purpose. Being aware of the methodological developments of recent historiography has brought a critical dimension to research against any claims of universal history or of objectivity in reading or writing history (Iggers 1997). Many historians have engaged in the discovery of group identities, trying to recover those voices that have been absent from history (Martínez Vázquez 2013).

Peter Burke refers to this way of writing history as the new history, or total history, written against the 'old' paradigm (Rankean interpretation), which dominated historiography in the past (Burke 1991). Together with social and cultural approaches, the article uses reception historiography to understand how journalists interpreted the apparition of the Virgin through the narratives of the children and how the masses received and interpreted the literary production of the journalists covering the events taking place in barrio Rincón (Jauss 1974).

The article focuses on popular Catholicism in Puerto Rico, in particular how Puerto Ricans experienced religion through the divine apparition of the Virgin at a spring in barrio Rincón of Sabana Grande in 1953. The article first provides a historical background to understand the Puerto Rico of the 1950s. Second, it describes the apparitions of the Virgin to eleven children in barrio Rincón for thirty-three days and responses to this event by government officials, the masses, and the official Roman Catholic Church. Third, the article describes how the children became intermediaries between the human and the divine through their functions as seers, prophets, and priests. Fourth, it considers and analyzes the correlation of Spiritism and Roman Catholicism in terms of the forms of religiosity operating in barrio Rincón, especially the devotions of the parents of some of the seers. Fifth, it describes and analyzes how water became a source of faith for thousands of people in search of meaning in their lives and of healing of their bodies and souls at a time when the island was going through rapid social changes. The article concludes by showing that the events of 1953 had ramifications that endure to this very day, despite the official position of the church, which continues to deny the apparition.

## 2. A Short Panorama of Puerto Rico before the Apparition of the Virgin of Rincón

According to Manuel Maldonado Denis, the United States envisioned Puerto Rico as one of its new territories, part of its attempt to expand its markets and control the Caribbean militarily (Maldonado Denis 1988, p. 65). Puerto Rico thus became the first colony of the United States, initiating a process in which the island became an object of the political, economic, cultural, and ideological apparatus of the United States. Maldonano Denis pointed out "The first four decades of imperial domination in Puerto Rico marked a process in which the island [was] dominated by industrial and financier capitalism of the United States" (Maldonado Denis 1988, p. 69) Sugar cane had become the major industry in Puerto Rico, and by 1930, the country increased its production massively, from cultivating 70,000 acres in 1898 to 250,000 acres in 1930, pushed by the expectations of North American shareholders. Coffee production had already collapsed as Puerto Rico had been prevented from exporting coffee to Europe and competing with other nations such as Brazil at lower prices, and tobacco production soon was monopolized by a US trust named Porto Rico American Tobacco Company. When the US stock market crashed in October 1929, the depression of the capitalist system unsettled Puerto Rico and led to desperate poverty. All the major industries were shaken to their core: sugar, tobacco, and needlework. Even the political parties were affected. In the 1930s, poverty became an acute problem for the Puerto Rican masses.

At the same time as Puerto Rico experienced the Great Depression—on steroids, as it were—nationalistic sentiments were flourishing in the island thanks to the Nationalist Party and a young radical named Pedro Albizu Campos (Ferrao 1990). By the time the Great Depression reached its climax in 1933, nearly half of the banks had failed, and the rest were not lending money. To make matters worse, the banks that controlled Puerto Rico's sugar cane industry cut wages all over the island. Industry after industry experienced the devastation of the Great Depression, and by the end of 1933, eighty-five strikes had erupted in the tobacco, sugar, needlework, and transportation industries (Denis 2015, pp. 116–17). In January 1934, the entire sugar industry went on strike and demanded payment of their wages. In their desperation, these workers asked the new leader of the Nationalist Party, Pedro Albizu Campos, to represent them in court. Albizu Campos was a bright *mestizo* who had earned an education at Harvard College and Harvard Law School, the first Latin

American to attend the prestigious university. On 23 February 1934, Albizu Campos led workers to victory and the sugar strike was settled, with workers accepting a better wage for a twelve-hour workday. After the apparent victory for the sugar workers, things went downhill for Albizu Campos, as the colonial (US) empire targeted him as a subversive. Due to the repression that the party experienced in the 1930s, by the 1940s, its popularity and political capital were in decline.

By the late 1940s, the political situation in Puerto Rico was changing, and people were beginning to accept the union with North America. In 1937, Luis Muñoz Marín was expelled from the Liberal Party because of his independent lineage and a year later he founded the Partido Popular Democratic (PPD), which campaigned for independence and social justice. In the elections of 1940, the PPD gained the majority votes and Marín was elected president of the senate. Once in the senate, his independent ideals waned, and in 1943, he advocated for an autonomous solution for Puerto Rico but not complete independence. According to Ayala and Bernabe, "Muñoz Marin proposed that the government buy the properties of the United Porto Rico Sugar Company, which included five sugar mills and 30,000 *cuerdas* of land. The poorer land would be withdrawn from sugar production, divided into small farms, and distributed to the landless" (Ayala and Bernabe 2007). In 1948, the United States conceded that Puerto Rico should elect its own governor and delegated all the executive powers of administration to the island. Muñoz Marin won the election and became the first elected Puerto Rican governor.

Sociologist Luis O. Zayas Michelli argued that from 1952 to 1960, Puerto Rico lived one of the most dramatic moments in its history when the sacred became profane and the profane became sacred due to the political activism of the Roman Catholic Church that embraced an open war for the soul of the island against the reformist views of Muñoz Marin's PPD by forming its own political party, the Partido Acción Cristiana (Micheli 1990, p. 175). With the United States' colonization of Puerto Rico, a process of secularization and liberalism began to influence the political and religious consciousness of the masses. The process of Americanization on the island shattered all the realms of social life in which the Roman Catholic Church had dominated for four hundred years. Among them, none was more pressing for Roman Catholics than religious education in the newly established public school system and reproductive rights of contraception and women's sterilization to control population growth (Silva Gotay 1991).

In short, the apparition of the Virgin of Rincón overlapped with a political conflict between the Catholic Church and the ELA, an economic crisis in the agricultural sector, the industrialization of manufacturing processes, and the creation of new cities. The spiritual experience for rural dwellers as the protagonists of the apparition of the Virgin in barrio Rincón demonstrated that the masses did not need the clergy as intermediaries with the divine, especially given that the clergy denied the veracity of the apparition, and indeed it solidified the devotional practices of popular Catholicism among the faithful.

### 3. The Virgin of Rincón and Human Agency

On 23 April 1953, the Virgin appeared to a seven-year-old boy named Juan Angel Collado in the village of Rincón in the town of Sabana Grande. This rural community had neither electricity nor a good public water supply. The small school had only one room, in which children from first to third grade learned together under the auspices of one teacher named Josefa Riós. On that 23 April, the school child in charge of getting water from the well was sick and could not perform his duty, so a woman tasked with cooking lunch for the children at school asked Juan Angel Collado and another boy named José del Carmen Rodriguez to fetch the water in his stead. The first written account of the events appeared in the newspaper *El Mundo* four days later when Samuel Irizarry, a journalist, interviewed Juan Angel. Recounted Juan,

> When I was standing near the ravine, I saw a whirlpool that moved the cane straws that were there. Suddenly I saw that the image of the virgin, dressed in white, emerged in color from the whirlpool. When I saw her, I stayed still

looking at her. The figure grew until it reached the branches of the mango tree there. When I returned to school, I told my classmates and they ran to see her without being afraid: Santia Rodriguez Lugo, Berta Pinto Lugo, Isidra Belen Moreno, Margarita Baez Ramos, Lydia Ester Santiago, Milagros Borelli Rivera, Leonor Galindo Alicea, Juan Rios, Milagros Rodriguez, Rosa Enid Rodriguez, and Ramonita Belén. All the children confirmed seeing the virgin. (Irizarry 1953e)

*El Mundo* newspaper was the major evangelistic source spreading the apparition of the Virgin, even though the journalists who covered the story remained skeptical of the 'supposed' apparition. From 27 April to 26 May, the apparition of the Virgin was front page news in *El Mundo*, with more than fifty articles written over that period of a month about the events taking place in barrio Rincón (Román 2007).

Although hundreds of people were flooding to barrio Rincón, many others were skeptical and blamed the parents of the children for the events. For example, people from the town blamed Dolores Pinto Torres, father of Bertita Pinto, for inventing the story for personal lucrative ends, as he owned the only store in barrio Rincón. When Samuel Irizarry, the major journalist for *El Mundo,* interviewed him and confronted him with the rumors in town of inventing the apparition, he adamantly denied the accusation by asserting his Christian character, that he went to church every Sunday, prayed twice a day, taught his daughter the doctrines of God, and would never lie about something so sacred (Irizarry 1953j). Another source of consternation was the inconsistencies and contradictions in the seers' descriptions of the Virgin. For example, the second written description by Bertita Pinto Camacho appeared on 1 May in *El Mundo*, in which she said, "In addition to the rosary, she carried a cross that shone and at her feet there was a dove and at that moment an angel came in a round cloud. The Virgin changed from a gray suit to a white one and she left with the angel, ascending slowly" (Carlo 1953). On Saturday, 2 May, Adrian Nelson Rivera, who was an art student in graphic design, drew a portrait of the Virgin following the description by an unnamed child: "To the right was the mango tree, below was the path that leads to the well, she had a crown with twelve stars, a cross, a rosary, above a cloud, and a dove within the cloud" (Irizarry 1953m). On 4 May, the teacher Josefa Ríos showed them a portrait of the Virgin of Fatima, and they corroborated that it was Fatima indeed (Irizarry 1953c). However, others perceived that it was the Virgin of Lourdes who appeared to Bernadette Soubirous in 1844 in the foothills of the Pyrenees. There were many similarities between Lourdes and the Virgin of Rincón. Their apparitions were in out-of-the-way places, both appeared above streams of water, were beautiful, floating over a cloud, smiling, with rosaries, clothed with a white robe, a veil on their heads, and holding a cross (Grovas 1953b). These similar patterns could represent a literary genre of its own or a topos that sheds light about continued patterns in the divine apparitions. Perhaps this is why they shared a similar structure in the descriptions of the apparitions. The final account describing the Virgin in 1953 came from Juan Angel Collado. He maintained that she had only seven stars, three on each side and a big one in the front, and a shining aura over her head (Santiago Sosa 1953). By this time, only seven of the eleven children were actively promoting the apparition of the Virgin, and even among themselves there were accusations and counteraccusations of who really had or had not seen the Holy Mother.

Margarita Báez informed journalist Rafael Santiago that the Virgin was not accompanying Juan Angel Collado, Santia Martinez Lugo, and Bertita Pinto in their processions to town. However, several days before, it was Isidra Martinez who accused Santia Mendez Lugo of lying to people about seeing the Virgin because the Virgin allegedly told Isidra that Santia could not see her (Irizarry 1953i). When the journalist went to school to interview Santia, she was crying because of the accusations and indicated that even before the apparitions, Isidra was always fighting with her. In response, Isidra told Santia "If you say that the Virgin did not tell me that you cannot see her, you are a sinner. What you need to do is to ask the Virgin for forgiveness." Thirty-six years after the events of 1953, there were only three seers with a common description of the Virgin and a different narrative of how things developed. In this new narrative, Juan Angel Collado did not invite the

whole group of children the same day but waited one day to tell the class what he had witnessed the day before. In this new narrative, Ramonita Belén, Isidra Belén, and Juan Angel Collado were the only children who saw the Virgin, and Bertita Pinto, Margarita Báez, Alba Garcia Galindo, Santia Mendez Lugo, and Milagros Borreli disappeared from the narrative. Their description of the Virgin of the Rosary, as it is known today, was of "a beautiful young lady, a beautiful face, hair with brownish curls, her skin was soft, neither white nor brown. She had like a tan. She was wearing a white suit, a brooch that buttoned, a blue cloak over her head, a belt around her waist. In her hands she carried a rosary, on her head a crown of seven stars" (Méndez de Guzmán 1989, p. 41). These discrepancies among the children, even though real, did not stop people from believing that somehow the Virgin appeared in barrio Rincón. Perhaps this is one evident example of how popular religion could manipulate the masses and serve as an alienating form of religion that does not liberate but rather leads to a superstitious Catholicism. These discrepancies among the children were real indicators that the story of the apparition could had been an invention by the families of the children. Among the key skeptics of the apparition of the Virgin were the clergy of the Roman Catholic Church.

The parish priest of Sabana Grande, Romualdo Ortiz, never visited the site of the apparition. Four days after the first apparition, on 27 April, people were inviting him to bless the site, but he declined and said that he would not go until he received permission from the church authorities. On 8 May, the children were trying to convince him to visit the well because if he did, "the Virgin will allow him to see Her hand." He never received permission to visit the site, as Monseñor Vicente Murga, rector of the Pontifical University of Puerto Rico at the time, did not want to legitimate the apparition (Irizarry 1953o). Not only did he not visit the site, but he also closed the doors of the cathedral on two occasions when Juan Angel Collado led processions. On one occasion, the masses wanted to enter, by force, if necessary, but the presence of police officers deterred them. On another occasion, Father Ortiz had to cancel the procession of the patron saint of the town, San Isidro Labrador, because the roads and streets of the town were impassable due to traffic, as the whole town was full of cars (Irizarry 1953h). Not surprisingly, priests from other dioceses were curious about the events taking place. For example, Fathers Torres and Toro, from the church from Salinas and Father Aponte from the church in Maricao visited the site and interviewed the girls who had allegedly seen the Virgin, but there is not record about these priests' views on the apparition or whether and how it affected their own spirituality (Irizarry 1953o).

Monseñor Rafael Grovas, from the Diocese of Ponce, argued that before Christians could accept the apparition of the Virgin as real, the events of the apparition must be submitted to the ecclesiastical authorities and its veracity be rigorously examined. For him, there were too many contradictions in the various accounts of the apparition to be taken seriously (Grovas 1953a). Finally, Bishop James McManus of the Diocese of Ponce did not inform the Vatican about the apparitions because he thought it was a sham invented by the children to get attention because they did not make any serious allegations about the apparition. Indeed, he thought that it was a mockery to suggest that the Virgin was sitting in a Jeep resting as one of the seers alleged, for her to lead processions to the church and for her not to give any messages to the faithful (McManus 1953). Even though church authorities wanted to delegitimize the apparition of the Virgin, their main goal was the submission of the Puerto Rican faithful to the authority of the church and not to the authority of children who experienced the appearance, as the children were functioning as seers, prophets, and priests.

## 4. Seers, Prophets, and Priests

Only the children allegedly witnessed the miraculous apparition of the Virgin Mary. No adult saw the Virgin in the thirty-three days of apparitions. As witnesses, they were representatives of divine revelation to the people of Rincón, Puerto Rico, and other countries, as news of the apparition traveled to New York, Mexico, the Dominican Republic, and Panama. On 30 April, because the children were no longer allowed to visit the well,

the Virgin continued her revelations in the classroom. The seers alleged that they saw the Virgin walking from the well to the school Lola Rodriguez de Tio at lunch break and then waiting for the teacher to open the door of the classroom. Once inside the school, the Virgin changed the daily routine of math, science, and history into a divine liturgy. She instructed the children to pray the rosary and requested the hymn *"Los Angeles Cantan"*. After the song ended, the seers alleged that the Virgin separated Bertita Pinto Camacho and Juan Angel Collado from the class, putting her left hand over Bertita and her right hand over Juan Angel, perhaps showing that they were the leaders or main seers of the group. The Virgin also asked the children to bring flowers to her, and when she touched them, the flowers withered and died. This was apparently supposed to be a miraculous sign for the teacher and the community. According to Dolores Pinto, father of Bertita Pinto, the appearance of the Virgin and the miracles she was performing were signs for people to repent and return to God (Irizarry 1953l).

The daily teaching activities at the school came to a halt, as the classroom took on a new meaning with the Virgin instructing the children in spirituality and devotion. As news of the apparition continued to spread, hundreds of people gathered outside the school to hear the children pray the rosary and sing spiritual songs, so much so that the supervisor for the Department of Education, Ramonita Soltero, recommended to her superiors that the school be closed due to the nature of the events taking place (Irizarry 1953d). The classes were also disrupted by the children leading pilgrimages to the spring, with hundreds of people following them in hopes of seeing the Virgin or a miracle. The processions were not only to the well: Juan Angel Collado, Bertita Pinto, and Isidra Belen also led pilgrimages to the church in the town of Sabana Grande, a three-mile walk from the school, with hundreds of people following them praying the rosary and singing to the Virgin. Father Ortiz ignored these processions; the church authorities prohibited him from legitimizing the apparition. Yet the more the church authorities told the masses not to follow the children on such pilgrimages and not to believe in the apparition of the Virgin, the more the masses continued their devotions and ignored the directives of the Roman Catholic priestly establishment. The pilgrimages to the shrine of the spring waters of barrio Rincón offered believers a privilege path to encounter God. As Jorge Mario Bergoglio stated in his lecture on "culture and popular piety" in Aparecida, "Popular piety has spaces for special expressions of faith through shrines. . .The shrine is the sign of the divine presence, the place of the ever-fresh renewal of the covenant of men with the Eternal and each other. . .The shrine is the place of the Spirit, because it is the place where the faithfulness of God reaches men and transforms them" (Bergoglio 2020, p. 257). Pilgrims traveled for hours to get to the shrine of the Virgin of Rincón and once there, had to wait in line for hours, sometimes from six to ten hours. During that time, pilgrims talked to each other, confessed their preoccupations about the transitioning from an agricultural society to an industrialized society led by modernism, prayed the rosary, and signed to the Lord. In other words, the shrine became a sacramental space of encounter with God and neighbor for pilgrims that poured their hearts to the Lord in the anticipation that God would be there to receive them with open arms. Bergoglio pointed out, "The pilgrimage is an expression of popular piety linked to the shrine. It has a profound symbolic expression that deeply reveals the human search for meaning and for encounter with the other in the experience of fulness, of what transcends us and is beyond all possibility, difference, and time" (Bergoglio 2020, p. 257).

The children were not only seers and pilgrimage leaders, but they also uttered several prophecies that contributed to people's hopeful expectations about the power of the Virgin to cure deceases, find soldiers lost during the Korean War, and even to find the body of a soldier who had disappeared in a river. Santia Martinez gave the first of these utterances. On 1 May, after the children had prayed the rosary and sung to the Virgin, the Virgin communicated to Santia Martinez that she would appear on 25 May for everybody to see her and to confer all kinds of favors and miracles (Carlo 1953). This prophecy that the Virgin would appear for everyone to see was an enormous sign of hope for thousands of people who were expecting or hoping for a miracle in their lives. People with all kinds of

diseases flooded barrio Rincón daily, and ever since that day, people travelled to the well from all parts of the island, the mainland, and other Latin American countries in hopes of a miracle.

Not only were the children seers and prophets, but they also had a priestly function in the daily activities taking place at barrio Rincón. According to Edwin Oliver James, "The primary role of the priest is that of the ritual expert, the one who has a special and sometimes secret knowledge of the techniques of worship, including incantations, prayers, sacrificial acts, songs, and other acts that are believed to bridge the separation between the divine or sacred and the profane realms" (James 2023). There are several examples of the priestly function of some of the seers of the Virgin at Rincón. Even though the seers argued that it was the Virgin who was teaching them in the classroom how to pray the rosary and sing songs, in reality the seers were children of devout Roman Catholics. As devout Roman Catholics, they regularly saw and experienced the liturgical celebrations led by Father Romualdo Ortiz in the Cathedral of Sabana Grande. They had seen Father Ortiz pray for the sick, baptize infants, bless people, and lead the liturgical celebration. As such, it is no surprise that the seers took upon themselves—or copied—priestly functions they had previously witnessed regularly.

Bertita Pinto and Juan Angel Collado were anointed by the imposition of the Virgin's hands over their heads, a ritual action that in Christianity represents the calling of a person to a special vocation within Christianity (Irizarry 1953k). Juan Angel Collado and Bertita Pinto were the primary agents in the daily processions and pilgrimages to the well, as well as to the Cathedral in Sabana Grande. In these processions, hundreds of faithful Christians followed them while praying the rosary and singing spiritual songs (Irizarry 1953d). People swarmed the house of Milagros Borreli to receive blessings by the seer, bringing to her water from the well, rosaries, scapulars, and other religious objects, such as portraits of the Sacred Heart of Jesus, the Virgin, and other saints. Even the police officers who were guarding the house of the seer asked Milagros Borreli to bless their badges (Irizarry 1953n). The seers Maria Borreli, Alba Galindo, and Isidra Belen Moreno even performed baptisms using the 'sanctified water' from the well that was blessed by the Virgin when they baptized two newborn babies in the local hospital of Sabana Grande (Irizarry 1953a).

One final example was the healing miracle of the son of Dr. Julio Montalvo, who was suffering from lung problems and could not breathe properly. Even after visiting Dr. Blanco in the town of Lajas, the seven-month-old boy's breathing did not improve, and his fever persisted. The boy's mother, Esperanza Olivera de Montalvo, who was Mexican and a devout follower of the Virgin of Guadalupe, took the boy to Rincón and visited the celebrated well where Juan Angel Collado was praying. She knelt beside Juan Angel and told him about her son's sickness. Juan Angel told her not to worry because the Virgin was caressing the head of the boy, and that the boy would be fine. He asked for "sanctified water" and made the sign of the cross on the boy's forehead (Irizarry 1953b). In the Roman Catholic tradition, upon entering the sanctuary the faithful, dip their fingers into holy water and make the sign of the cross then genuflect before entering a pew. Making the sign of the cross refers to the redemptive and saving act of Jesus in his death on the cross for humanity. The cross is a reminder of the paschal mystery of salvation in the passion, suffering, death, resurrection, and ascension of Jesus Christ. The sanctified water is also symbolic of the baptismal waters of initiation in the faith. In this sense, the young seers of Rincón were functioning as priestly agents through liturgical acts of worship, performing baptisms, and even giving Communion.

## 5. Spiritism and Popular Catholicism

Not everyone was as welcoming of the children as this mother. There were certainly people who interpreted the seers as spiritist mediums. In the 1920s, an adolescent girl named Julia Vázquez caused a great commotion on the island with her healing ministry as a devout *espiritista*. As in the case of the calling of shamans in indigenous religions, Vázquez went through a painful sickness during which she received many visions or revelations

([Siikala 1987](), p. 210) She had visions of men wearing military uniforms marching in the mountains of Puerto Rico (World War I had just ended), saw sentences in a language she was unable to recognize that angels subsequently interpreted for her, and saw visions of crowns in heaven. In one vision, she saw thousands of people coming to her for healing through the curative water of a spring besides her house ([Rojas Daporta 1953b]()). Indeed, in the 1920s, thousands of sick people traveled to the town of San Lorenzo to receive healing at the hands of *la Samaritana*, as Julia Vázquez was known, and through the spring waters of the mountainous town of San Lorenzo. Historian Reinaldo Román points out, "The water that la Samaritana dispensed was procured from springs near her home and was later 'magnetized' while the *médica* was in communication with her guide, a spirit identified as none other than San Lorenzo's old parish priest" ([Román 2002](), p. 163).

In Puerto Rico of the early 1900s, Spiritism was both popular and widespread. Carlos Cardoza-Orlandi explains that Spiritism "is a French-diaspora religion that finds new effervescence as it cross-fertilizes with Roman Catholicism and other African-diaspora religions in the New World" ([Cardoza-Orlandi 2006](), p. 735). Spiritism is a syncretistic religious expression influenced by the writings of Alan Kardec, by the Bible—especially the Gospel of John—and by African-diaspora religion. Cardoza-Orlandi elaborates

> Spiritism was grounded on the belief that the spirits of dead people were capable of communicating with the material world, the spirits could be investigated scientifically, the spiritual realm has a hierarchical existence in which those close to the material world could be considered as evil, and spirits in this world were free and could achieve perfection through incarnations. ([Cardoza-Orlandi 2006](), p. 736)

As a religion of reciprocity, in which the spiritual and material world need each other, the Spiritist medium played a crucial role not only as the conduit or intermediary between the two worlds, but also in ensuring that the new religion would survive in the New World ([Bram 1972]()). According to Cardoza-Orlandi, "the medium exercises the pass which unlocked the pathways making the spirits available for the community gathered in a room of a house or an adjacent structure, around the table or, in more recent locations, a church-type building and space configuration" ([Cardoza-Orlandi 2006](), p. 736). The correlation of the child seers with mediums prompted considerable consternation among the people of Rincón, particularly when the first account of the vision or apparition that Juan Angel Collado reported was that he saw the spirit of a dead person, a statement he retracted the next day after a conversation with his grandmother ([Irizarry 1953f]()). Nonetheless, people in town believed that it had indeed been the spirit of a dead person, not least because a devout Roman Catholic teenage girl had earlier died at the same location on which the school was later built. Others claimed that a man was stabbed close to the well and that it was his spirit that the children saw and not the Virgin. Even today, the superstition persists in Puerto Rico that if a person dies prematurely or is a victim of an accident or a crime, such a person will not rest in peace and becomes a ghost. Perhaps Bishop McManus of the diocese of Ponce deliberately referred to the superstition ingrained in the Puerto Rican psyche to cast doubt on the alleged apparition of the Virgin and to attribute it instead to the ghost of a dead person ([Rivera 1953]()).

There were other indicators that the Roman Catholicism practiced by Juan Angel Collado's family was influenced by Spiritism, even though his grandmother vehemently denied that she had anything to do with the religion. When Samuel Irizarry, the primary journalist from *El Mundo* covering the events of barrio Rincón, interviewed María Reyes Camacho de Pinto in her house—the mother of Bertita Pinto—he noticed that in one corner of the small house there were several statues of the Virgin of Miracles, the Virgin of Carmen, the Virgin of the Rosary, the Sacred Heart of Jesus, and many small stamps of saints and photos of family members. At the back of the altar was a white silk cloth inscribed in blue pencil with the names Juan Angel, Berta, Ramonita, Margarita, Alba, Gloria, Leonor, Santia, Milagros, and Isidra, indicating that they had seen the Virgin. Below these names appear the names of Berta and Margarita and another note about Angelita who did not see the Virgin. The mother told the journalist that the Virgin visits her home because she is the

most Catholic person in the neighborhood. Her views of angels were another indication of spiritist influences. She had had sixteen children, but seven of them had died at a young age. She believed that because they were children, they became angels who accompanied the Virgin in her appearances. Not only did she believe that the seven angels she had seen were her sons, but she also believed that the angels had given their wings to the seven elected seers: Juan Angel, Bertita, Ramonita, Margarita, Alba, Santia, and Milagros (Irizarry 1953r). The Spiritism to which the rural population adhered was not the scientific discourse of Kardec adopted by the Puerto Rican elites, but rather a form of popular Spiritism that was attuned to popular Catholicism with its beliefs in ghosts, spirits, mediums, saints, virgins, and angels as either protectors or adversaries of humanity (Micheli 1990).

Perhaps the influence of María Reyes Camacho de Pinto on the narrative of the apparitions could explain why the number of seers changed from eleven at the beginning of the apparition to these seven seers. That was, moreover, not the only instance of such alleged sightings. For example, her daughter Bertita Pinto told Irizarry in another interview that once when she was going to her house, an angel was following and protecting her, and when she went to Isidra's house, she saw two angels dressed in blue there (Irizarry 1953q). Juan Angel had been sick in his house for three days due to exhaustion and, according to Bertita and Isidra, the Virgin had allegedly told them to go to his house and that they would find her there. Once in the house, Juan Angel said that the Virgin visited him with two angels, and one angel gave him an injection under his ear and another in his right arm, making him feel better and ready to resume his ministry (Irizarry 1953g). Not only did the children see the Virgin and angels, on one occasion they even reported that the devil had appeared to them. According to Ramonita Belen Otero, the devil visited the school as a punishment because Margarita Báez had disobeyed the Virgin. Ramonita described Lucifer as "a skinny man dressed in black, covered with a cloak. His hands were long and skinny, and he had very long nails, his face was long and ugly with two horns on his head" (I. Irizarry 1953). The depictions of good and evil in rural communities, as expressed by the seers, were common in the collective imagination of rural dwellers. Taino, African, and Spiritist religions had commonalities to biblical accounts of demons tormenting human beings, exchanges between humans and spirits to prosper or to harm others, and magical incantations using water to heal the sick, which were very popular to the Spanish Catholicism brought to the 'new world' (Vidal 1989, p. 155).

The importance of water to overcome evil spells was part of rituals of deliverance in Puerto Rican folk religion. Vidal points out "There are those who believe that to overcome a spell by a witch the best thing is to take three sea baths for three consecutive days and taking three sips of salt water on each occasion. Another common resource was called *agua de espanto*, which was prepared by leaving the leaves of various odorous plants on sundown for three consecutive nights. On the third day they are boiled in water, and, when the infusion cools, it is poured to the head of the victim" (Vidal 1989, p. 155). It is important to notice that water was never used by witches in the popular imagination of Puerto Ricans to cause evil or to harm enemies through enchantments. Instead, water was always used as a medium for deliverance of evil and to bring goodness or prosperity.

## 6. The Virgin and Miraculous Water

In almost all religions, water is symbolic of creation, cleansing, sanctification, and new life. The first verses of the creation story of the Hebrew Scriptures recount that in the beginning, God created the heavens and the earth and that the earth was formless and empty, darkness was over the surface of the deep, and the Spirit of God was hovering over the waters. In the liberation of the Israelites from the yoke of the Egyptians, Moses led the people to freedom by parting the waters of the Red Sea for them to start their journey to the promised land. Abraham, the patriarch of the Jewish people, dug several wells on his journey to the land God promised him, and it was likewise at a well in the desert that Hagar received sustenance for her, and Ishmael after God opened her eyes to see a well of water. In the New Testament, no other story of water is more moving than the encounter of

Jesus with the Samaritan woman at Jacob's well. In a theological exchange regarding the correct place for worship, Jesus tells the Samaritan woman that he will give her water that will never dry up and that she will never be thirsty again because such water will become a spring of living water leading to eternal life.

In African traditional religions, creation narratives about the beginning abound with water. In the creation story of the Yoruba of Nigeria, all that existed in the beginning was water, land, and sky, ruled over by Olorun. In the creation myth of the Kuba people of what today is known as the Democratic Republic of Congo, Mbobo created out of nothing when he was alone, and darkness and the primordial water covered the earth. In ancient Taino religions in the Caribbean, the birth of the ocean and the life within was the first act of creation by the Goddess Abatey, who created herself out of nothing. In the Japanese creation myth, Izanami and Izanagi created the ocean and stirred it with a spear that, when crystalized into drops, created the islands. Water is also divinized in several religions. In Hinduism, the Ganges River is associated with the divine, as it was allegedly conceived as a goddess to purify humanity. In Cuban Santeria, Yemaya is the Orisha of the ocean and Ochun is the Orisha of the rivers, fertility, and sensuality. In Mahayana Buddhism, Kwan Yin is known as the Water-Moon Bodhisattva of compassion, who was born from the waters of the Lotus flower.

The Virgin of Rincón appeared to Juan Angel Collado first and later to eleven of his classmates at the waters of the spring close to the school of Lola Rodriguez de Tió. All the miraculous healings happened at the spring or because people drank or washed themselves in water from the spring. The first recorded miracle happened on 5 May when a young woman named Margarita Pallares was healed after spending ten days in the Presbyterian hospital of Santurce without improvement in her health. Her father, Alejandro Pallares, traveled to Rincón expecting a miracle from the Virgin and, after waiting in line for four hours, when they finally arrived at the spring, Alejandro and his wife took water from the spring and washed Margarita's chest, stomach, and head with it. After drinking water, the family knelt in front of the spring and began to praise the Virgin for healing the young woman (Irizarry 1953p). The next day, an old man named Jose Matos Rivera from the town of San German alleged that after drinking the water and washing his feet with it, he was finally able to walk again without his walking stick (Irizarry 1953d). María Luisa Irizarry from the town of Lajas was the third person to receive healing from the waters of the spring. According to María, she had suffered severe stomach pain for three months and had been unable to eat solid food. After drinking water from the spring and washing her stomach with the water, she was instantly healed and was able to eat solid food once more (Irizarry 1953o).

After these healings, thousands flooded to the spring in Rincón seeking comfort and relief from their ailments. Many of these people had not been helped by modern medicine and felt they had no alternative but to put their lives in the hands of the merciful Virgin. People began to bring to the spring relatives who had been bedridden for months or years, people suffering from dementia, people who were lame, blind, paralyzed, and all kinds of other illnesses, hoping against hope that they would be cured of their maladies. Not only were people healed of physical maladies, but some attributed to the spring mundane things such as finding a long-lost check. This is what happened to a woman employed by the Bank of San Juan who had lost a check for USD 250,000 and who did not want to be identified by name. After days spent looking for the check, the devout Roman Catholic decided to travel to Rincón, expecting a miracle. She went to the spring and filled up a bottle of water. Once she returned to the bank, she sprayed the water on the four corners of the office and began to pray the rosary, when suddenly she found the check in a pile of papers, which she had already examined more than ten times without finding anything. She attributed finding the check to the powers of the spring waters (I. Irizarry 1953).

One of the most commented on miracles was that of Georgina Politis de Rivera, better known as "the Greek woman." She was thirty-eight years old, but her sufferings made her look as if she were eighty. She was apparently a schizophrenic, and one of the shock treatments that had been administered to her in that era fractured her cervical vertebrae,

leaving her paralyzed. She lived in Miami but visited Puerto Rico frequently, where an osteopath named Dr. F.E. Mundo treated her, prescribing a cervical collar to prevent any movement. When she learned about the miraculous healings in Rincón, she decided to travel there from Miami, hoping for a miracle. As a Greek Orthodox believer, she thought the Virgin Mary—the *Theotokos*—would hear her prayer for help and heal her through the waters of the spring. Her miraculous healing was the only one corroborated by a doctor—the one who had treated her for years. He indicated that only a miracle was able to heal her (Rojas Daporta 1953a).

For Julia Vazquez, known as "la Samaritana healer" of the 1920s, the miracles of Rincón were not so much about the Virgin, but rather about the miraculous waters of Puerto Rico's springs. As an *espiritista*, she believed it was the people's faith in the waters that healed them, not the Virgin. In any case, Puerto Ricans of all kinds connected the special waters of the spring in Rincón with the divine appearance of the Virgin, linking them to their primordial past and making a space in their lives for both the divine and the mundane amid rapid social changes. The era of industrialization and a new (commonwealth) type of government that had disrupted the normal flow of life in rural areas created insecurities about their future, and popular Catholicism expressed in this instance through visions and healing helped *jibaros* cope with this new reality. However, "la samaritana" was talking as a spiritist and not as a devout Roman Catholic. The spring waters of barrio Rincón were sanctified by the Virgin. As the mother of Jesus, the Virgin "is the Lord's perfect disciple". As the final document of the Fifth Conference of Bishops in Latin America and the Caribbean attested, "As the Father's interlocutor in his project of sending his Word to the world for human salvation, Mary, by her faith, becomes the first member of the community of believers in Christ, and also collaborates in the spiritual rebirth of the disciples". The Virgin's apparition at barrio Rincón was a spiritual reawakening for the people of Puerto Rico, not only for Catholics, for the whole spiritual religious revival of the island, as she draws multitudes to strengthen fraternal bonds among all.

In the New Testament, no other gospel presents the topic of water more than John. In the story of the Samaritan women at Jacob's well, Jesus asked the woman for water and in the exchange told her about his identity, "If you knew the gift of God and who it is that asks you for a drink, you would have asked him, and he would have given you rivers of living water". Of course, the gift of God was Jesus Christ and his salvific action on the cross of calvary to take away the sin of the world. The pilgrims walking and patiently waiting for hours at the spring of barrio Rincón understood that the waters they were approaching and placing their faith on was a spring of living waters sanctified by the Spirit of God. This is why there were so many testimonies of people being healed of illnesses of all types, as people who were paralyzed walked, those who were blind recovered their sight, and sinners were restored by grace to the community of believers because those who drank from the water would have in themselves a well of water leaping to eternal life. In this sense, the Virgin pointed to the redemptive reality of Jesus through the grace of God manifested in the Holy Spirit. The spring waters of barrio Rincón became a sacramental space in which the whole process of salvation was manifested in miraculous healings of the body, but more importantly, many souls were healed from their sins.

## 7. Conclusions

The apparition of the Virgin in barrio Rincón is a classic example of popular Catholicism. On the one hand, the official priestly functionaries were concentrated in the metropolis where they ministered to the elites and urban dwellers. The official Catholicism they represented and dispensed was rooted in the sacraments, the liturgy, and the homily. It was a Catholicism imported from Spain and which served to control the colony and its elites. Due to the long distances from the city, priests barely ministered to the rural population. This lack of pastoral catechism contributed to popular expressions of Catholicism, rooted in and born of the imagination of its adherents. On the other hand, there was popular Catholicism or the Catholicism of the rural masses. This popular Catholicism was not

sacramental, liturgical, or homiletical, but it was in continuity with such practices. The sacramental nature of the ministry of the children of barrio Rincón—in baptizing, anointing, leading processions, and speaking in the name of the Virgin—were representative of the official teachings of the church. However, their ministries were filtered by the rural interpretations that they learned from their parents, grandparents, and the collective memory of rural accounts of saints, ghosts, angels, the devil, and the Virgin. As Robert Schreiter states, "Popular Religion no longer needs to be dismissed as deviations brought on by psychological need or lack of proper evangelization. It can be seen as an authentic way of living out the message of the Gospel" (Schreiter 1999). This is what professor Orlando Espín at the University of San Diego calls the Sensus fidelium or the living witness and faith of Christians in popular Catholicism. Espín points out, "it is this 'faith-full' intuition that makes real Christian people *sense* that something is true or not vis-à-vis the gospel, or if something is acting in accordance with the Christian gospel of not, or that something important in Christianity is not being heard" (Espín 1995, p. 150). Espín's arguments about the Sensus fidelium does not mean that all aspects of popular piety are valid or necessarily right. Nonetheless, "because the foundational origin of the *sensus fidelium* is the Holy Spirit, it can be said that this sense of the faithful is infallible, preserved by the Spirit from error in matters necessary to revelation" (Espín 1995, p. 151). In other words, the Holy Spirit would never contradict the final revelation of God in Jesus Christ given in the Scriptures. In the apostolic exhortation *Evangelii Gaudium*, Pope Francis pointed out

> In all the baptized, from first to last, the sanctifying power of the Spirit is at work, impelling us to evangelization. The people of God is holy thanks to this anointing, which makes it infallible *in credendo*. This means that it does not err in faith, even though it may not find words to explain that faith. The Spirit guides it in truth and leads it to salvation. As part of his mysterious love for humanity, God furnishes the totality of the faithful with an instinct of faith—*sensus fidei*—which helps them to discern what is truly of God. The presence of the Spirit gives Christians a certain connaturality with divine realities, and a wisdom which enables them to grasp those realities intuitively, even when they lack the wherewithal to give them precise expression. (Pope Francis 2013, #119)

Therefore, the apparition of the Virgin in Rincón is symbolic of the belief that the Puerto Rican people had in their deepest expression of the divine. Even when the ecclesiastical authorities denied the veracity of the alleged apparition of the Virgin, the Puerto Rican faithful continued to believe that the Virgin had indeed appeared to the children/seers. In a sense, this pushing back against the authority of Bishop Joseph McManus and against the official posture of the Catholic Church and conferring such authority on the children/seers as revelation from God was an instance of Puerto Rican people taking authority into their own hands. The more the Catholic Church denied the apparition and threatened the faithful with excommunication, the less the Puerto Rican faithful cared about what the official church had to say. For example, though Bishop McManus ordered all his priests on the island to read to their congregations an official pastoral letter on Sunday 24 May, the day before the prophecy that the Virgin would appear again for everyone to see, a letter discouraging Christians from traveling to barrio Rincón, it is estimated that over 100,000 people gathered the following day to see the miracle of the apparition. As the final document of Aparecida states, "The Church, like the Virgin Mary is mother. This Marian vision of the church is the best antidote to a merely functional or bureaucratic church" (Pope Francis 2013, #268). Even today, seventy-one years after that first apparition of the Virgin in barrio Rincón, the official position of the Catholic Church is that there was no apparition. This denial by the official ecclesiastical authorities has not stopped the faithful from gathering in their thousands every 25 May to offer their lives in service to the Virgin of Rincón. The apparition of the Virgin in Rincón in 1953 and the continuing devotion of the masses to her is an example of the Puerto Rican style of popular Catholicism.

**Funding:** This research received no external funding.

**Data Availability Statement:** Data available in public access repository. The original data presented in this study are openly available at https://gpa.eastview.com/crl/elmundo/ (accessed on 4 March 2024).

**Conflicts of Interest:** The author declares no conflict of interest.

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
