# Peer review of "Popular Catholicism Puerto Rican Style: The Virgin of Rincón, Human Agency, and Miracles"

_religions, doi:10.3390/rel15040463_

Round 1

Reviewer 1 Report

Comments and Suggestions for Authors

The topic, an apparition of the Virgen not recognized by the Catholic Church, is well researched. The different phases and the contradictions between the stories and visions is well detailed. The author has used adequately the sources to give a good description of the event.

The article lacks articulation between the different parts. The historical part does not relate well with the rest. The topic of the relationship with the clergy has to be developed.  

Method: You should have a paragraph on your method of research.  

Some problematic topics:

“Today, popular Catholicism is a source of liberation, cultural identity, and spirituality. The religious vitality of the poor and disfranchised in relation to their cultural patrimonies are new sources to construct theology from the grassroots and their daily life experiences”. Beginning of the article.

The author is too biased towards popular Catholicism. He manifests a lack of knowledge concerning the position of the Church on popular Catholicism and a lack of understanding of some of the most problematic aspects of popular Catholicism.

Popular Catholicism is not a source of liberation: in your text you have shown how some “messages” of the Virgin were used in a conflict among children. Some aspects of Popular Catholicism may alienate people… In this case, you describe a use of visions and messages to gain power (maybe money) over devoted Catholics.

The clergy has always been ambivalent with popular Catholicism. Surely enough, they, today, do not shun (in general) popular Catholicism but they consider that it has to be purified, see the final document of Aparecida.

You have to verify the meaning of Sensus Fidei. It does not mean that beliefs of the mass are necessarily right. Pope Francis has pondered the topic inspired by Juan Carlos Scannone (Theology of the People). Pope Frances wrote the chapters in the document of Aparecida on popular piety.

You mention syncretism but by itself, Catholicism, right from its origin, is syncretic, no need to look for spiritism. It can be argued that spiritism as developed by Alan Kardec, stems from science and Christianism... Even in the Bible, spirits are mentioned. Your reflection on the purifying power of water is too superficial.  

Missing in the article:

There is no mention in your article of other local apparitions of the Virgin. No mention of the possible relation between this apparition of the Virgin and other international apparitions of the Virgin. Your description recalls the Lourdes apparition, very popular worldwide. Maybe, the Porto Rican apparition has been reinterpreted in reference to Lourdes. You may even postulate a special religious gender: apparition of the Virgin with miracles, purifying water, children…

Comments on the Quality of English Language

minor editing will be required.

Author Response

Thank you very much for taking the time to review this manuscript. Please find the detailed responses below and the corresponding revisions/corrections highlighted/in track changes in the re-submitted files in red.

1) In the first draft I wanted to take a phenomenological approach and don't get tangled on the veracity or not of the apparition. In my revision I made explicit some of the problems of popular Catholicism as you pointed them.

2) Even though I agree that there are aspects of popular Catholicism that need to be purify, all forms of Christianity need purification. The reviewer took an absolutist position when you said "Popular Catholicism is not a source of liberation." I disagree with your assessment about how popular Catholicism is seeing today by the clergy, bishops, and even Popes. For example, even though Benedict XVI said that popular religiosity needs purification in some aspects, he also was very positive in his inaugural address in Aparecida. "The rich and profound popular religiosity, in which we see the soul of the Latin American peoples: -love for the suffering Christ, the God of compassion, pardon and reconciliation...the profound devotion to the most holy Virgin of Guadalupe, the Aparacida, the Virgin invoked under various national and local titles. The Aparecida document is positive about popular religion: "The Holy Father emphasized the rich and profound popular religiosity, in which he sees the soul of the Latin American peoples. He called for it to be protected and promoted because their piety manifest a thirst for God that only the simple and poor can know" (#258). "Among the expressions of this spirituality are: patron saint celebrations, novenas, rosaries, the way of the cross, processions, dances and songs of religious folklore, affection for the saints and angels, solemn promises, and family prayer (#259). "Popular piety delicately permeates the personal existence of each believer, and even though he/she lives in a multitude, it is not mass spirituality. At different moments of daily struggle, many go back to some small sign of God's love: a crucifix, a rosary, a candle lit to accompany a child in illness (#261). "Popular piety is an indispensable starting point in deepening the faith of the people and in bringing it to maturity...When he say that it has to be evangelized or purified, WE DO NOT MEAN THAT IT IS DEVOIT OF THE GOSPEL WEALTH. We simply want all members of the believing people, recognizing the testimony of Mary and also of the saints, to try to imitate them more each day (#262). "We cannot deprecate popular spirituality, or consider it a second mode of Christian life, for that would be to forget the primacy action of the Spirit and God's free initiative of love...It is a spirituality incarnated in the culture of the lowly, which is not thereby less spiritual, but it is so in another manner (#263). "Popular spirituality is a legitimate way of living the faith, a way of feeling part of the Church and a manner of being missionaries (#264).  

2) When Bergoglio used the term evangelization of cultures, he is not referring to popular Catholicism but rather to the 'culture of death' whose most evident signs are: an increase in poverty and in extreme poverty, concentration of wealth, lack of equality, law of markets, neoliberalism, financial paradises, a crisis of democracy, corruption, migration, social discrimination, terrorism, environmental pollution, family crisis, abortion, euthanasia, subjectivism, consumerism, imposition of modern culture and contempt for ancestral cultures, individualism, a crisis of values, moral relativism, placing a distance between faith and life" in "Unique Challenge: The Crisis of Civilization and Cultures Position Paper at Aparecida." Even though he said the same as Benedict in terms that there are aspects on popular Catholicism that need re-evangelization or purification, he was very positive of PC. For example, "The inculturation of faith is one of the priorities of the Church today. For this reason, popular piety is the faith of the simple people that becomes life and culture. It is the people's particular way of living and expressing their relationship with God, the Virgin and with the saints, in the private and intimate environment and also in community" (p.250). "To positive value popular piety, we have to start from a radically hopeful anthropology...The appreciation of popular piety starts from a conception of man as transcendent, sacred beings...Popular piety is simply the religious tendency of believing people who cannot but express publicly, with sincere and simple spontaneity, their Christian faith (p.250). "In the face of modernization, even more so of postmodernity, it has been a communitarian form of resistance (liberation), a PROPHETIC CRY of the person who does not want to deny the mystery and transcendent in the horizon of life (p.251). 

3) I expanded and clarify the sensus fadelium.

4) I said exactly what you are saying about spiritism in page 10 and expanded the topic of water using the gospel of John.

5) I don't see the need to bring the apparition of Monseratte that happened in the 16th century to this paper. There is no correlation from an event in the 16th century that was conferred canonical status in 1994 by John Paul II. Also, to reinterpret the Virgin of Rincon through Lourdes is a historical problem. This is why the Virgin of Rincon is recognized today as the Virgin of the Rosary or simply the Virgin of Rincon, it is a national Puerto Rican apparition, and not an European counterfeit.

Reviewer 2 Report

Comments and Suggestions for Authors

Abstract needs to be improved, e.g.specifically mention the theories discussed and findings (e.g. spiritism in popular syncretism vs official catholicism issues ).

While the study is more of a descriptive nature, some arguments seem to be implicit about the nature of this popular religion indicated as liberational and syncretistic. It is suggested to more clearly indicate these in the abstract and introduction.

"the first verses of the mythical creation story" Please reevaluate the use of "myth" and "mythical" in the ensuing paragraphs.

Spiritist or spiritualist? Please reevaluate the use of the terms interchangeably, or, clarify their interchangeable use, e.g. in "Spiritualist influences"

Comments on the Quality of English Language

“how water became a source of faith” instead of “how water becomes”

“on steroids, as it were, nationalistic…” (needs comma instead of full stop after “were”)

Author Response

Thank you very much for taking the time to review this manuscript. Please find the detailed responses below and the corresponding revisions/corrections highlighted/in track changes in the re-submitted files in blue.

1) I improved the abstract by providing the theories used in the paper and the findings of such theories.

2) I highlight the points you suggested and made explicit the problems of the receptors or masses with the clergy and their acceptance of popular religion.

3) I eliminated the language of myth.

4) I use only the term spiritism and spiritist instead of spiritualism.

5) I corrected the grammatical things you pointed out.

Round 2

Reviewer 1 Report

Comments and Suggestions for Authors

The article has been improved. It is an interesting topic as it stands .

Just a few changes should be made. 

Your comments on pope infallibility have to be erased. they are not necessary and they are not correct.

Your comments on the Samaritaine could be reduced, I do not think the bible is the direct reference. The reading of the bible amongst Catholics started as a consequence of Vatican II and maybe the growing influence of the Evangelicals.

By referring to Lourdes, I did not mean that Lourdes was used as a model. I was thinking of a topos or even a litterary genre. The Virgin sheds tears in many places around the world. In the same way, her apparitions are often associated with a spring and some sort of hydrotherapy. The idea is to connect this apparition in Puerto Rico with others. We are not judging the genuineness. We are observing possible patterns.

Your co 

Comments on the Quality of English Language

Minor editing may be required.

Author Response

Thank you for the comments.

1) I erased the sentences about the Pope's infallibility.

2) I cut half paragraph describing water in John 1-3. I also cut the lengthy quote of Pentecostal biblical scholar on John 4.

3) I included a paragraph in blue connecting Lourdes to Rincón and showing the patterns of both apparitions.